# Sorption of Selected Heavy Metals with Different Relative Concentrations in Industrial Effluent on Biochar from Human Faecal Products and Pine-Bark

**DOI:** 10.3390/ma12111768

**Published:** 2019-05-31

**Authors:** Koetlisi Andreas Koetlisi, Pardon Muchaonyerwa

**Affiliations:** Soil Science Discipline, School of Agricultural, Earth and Environmental Sciences, University of KwaZulu-Natal, Private Bag X01, Scottsville 3209, South Africa; Muchaonyerwa@ukzn.ac.za

**Keywords:** biochar, faeces, heavy metals, pyrolysis, sorption

## Abstract

The removal of heavy metals from effluents at source could reduce contamination of soil and water bodies. A batch sorption experiment was performed to determine the effects of feedstock of biochars pyrolysed at increasing temperature on sorption capacities of Cu, Cr and Zn from industrial effluent and aqueous solutions. Sewage sludge, latrine faecal waste and pine-bark biochars were used. The sorption data were fitted to the Langmuir isotherm. Maximum sorption capacities of latrine waste, sewage sludge and pine-bark biochar (350 °C) were, respectively, 313, 400 and 233 mg kg^−1^ for Zn, 102, 98.0 and 33.3 mg kg^−1^ for Cu, and 18.9, 13.8 and 67.1 mg kg^−1^ for Cr from industrial effluent. Conversely, sorption capacities from single metal solutions were 278, 227 and 104 mg Zn kg^−1^, 97.1, 137 and 21.3 mg Cu kg^−1^, 122, 106 and 147 mg Cr kg^−1^ on latrine waste, sewage sludge and pine-bark biochar, respectively. Step-wise regression analysis showed that the combined effects of ash, fixed C, pH influenced Zn sorption, ash and fixed C affected Cu sorption, and Cr sorption by ash and specific surface area of the biochar. The findings of the study imply that biochar from human faecal waste, particularly sewage sludge, has the potential to be utilized as sorbents of heavy metals from multiple metal effluent and that the sorption is affected by relative concentrations.

## 1. Introduction

Environmental contamination with heavy metals poses risks to the biosphere [1,2] and human health [3,4], with the soil-plant-human pathway being a primary mechanism of transfer. Anthropogenic activities such as mining, agriculture and other industrial activities result in elevated levels of heavy metals in soils [1,5,6]. For example, industries specializing in electroplating and production of batteries, fertilizers and pesticides produce wastewater, which often contains high concentrations of heavy metals [1,7,8]. The discharge of the effluents into sewer systems enriches sewage sludge with the metals, which presents challenges for land disposal of the sludge. The occurrence of multiple heavy metals, from different sources, and their non-biodegradability and persistence in the environment, exacerbate the disposal challenges and pose major ecological risks [9]. Treatment of industrial wastewater, at source, makes it safer for discharge into the environment and reduces environmental pollution [10,11].

Precipitation and complexation are known to be the most effective strategies in removal of metals from effluents but the requirement of specialized reagents and expensive equipment makes them too expensive [7,12,13]. Other available technologies for treatment of wastewater, including membrane filtration, ozonation, advanced oxidation and adsorption, have also shown high effectiveness [10,11,14], but the high costs also limit their application. Biochar has shown potential to be a cost-effective biosorbent for the removal of toxic metals from aqueous solutions and wastewater [6,13,15,16,17]. The effectiveness depends on biochar characteristics owing to properties of the feedstock and pyrolysis conditions [18,19,20,21].

Studies by [22,23,24] have shown the capacity of different biochar types to remove heavy metals such as Cd, Cu and Zn from aqueous solution. Biochar types from nut shells, plum stones, wheat straws and grape stalks and husks were highly effective in removal of Cd from aqueous solution, with wheat straw and grape husk and stalks being the most efficient [19]. Biochar produced at higher pyrolysis temperature has been shown to increase Cd sorption capacity due to higher aromatic structure, pH and surface area [15,25]. It is essential to note that most of the studies on the removal of metals by biochar were done with single metals [23,24] with fewer on mixtures of metals, at equal concentrations, in aqueous solutions. There is evidence that presence of competing metals in aqueous solution affect the sorption of individual metals. For example, the adsorption capacities of metals from aqueous solution onto sesame straw biochar were in the order of Pb > Cd > Cr > Cu > Zn from single-metal and Pb > Cu > Cr > Zn > Cd multi-metal solutions [26]. Where multiple metals were used, the mixtures were often prepared with equal concentrations [26]. However, in reality, the metals occur at different concentrations in industrial effluents, and the concentration effects may affect the preferential uptake of metals by sorbents.

In a study on sorption and desorption of heavy metal on biochar and commercial activated carbon, [27] reported that various operating factors such as adsorbent dose and initial concentration are influential on the sorption. In that study, sorption capacity increased with increase in initial concentration of metal ions [27]. There is need to understand the potential of using different types of biochar from locally available feedstocks on removal of multiple metals from a typical industrial effluent.

Biochar from human faecal wastes [28] differed in their physicochemical properties, with higher cadmium (Cd) sorption capacity, compared to those from pine-bark [29]. The Cd sorption generally declined with increase in pyrolysis temperature (350–650 °C) from 200 to 167 for latrine waste biochar, and from 143 to 36 mg kg^−1^ for sewage sludge biochar [28], while it slightly increased from 20 to 34 mg kg^−1^ for pine-bark biochar [29]. Whereas ash content and phosphorus composition explained the sorption of Cd on faecal biochar, fixed C, pH and Brunauer-Emmett-Teller (BET) surface area appeared to be more important in pine-bark biochars. The potential of these biochar types to remove mixtures of heavy metals from industrial effluents needs to be understood. Therefore, the objective of this study was to determine effects of biochar types from human faecal waste and pine-bark on sorption of multiple metals from industrial effluent and single metal aqueous solutions.

## 2. Results

### 2.1. Sorption of Zn from Multiple Metal Effluent and Single Metal Solutions

Based on the shape of the sorption isotherms, biochar from pine-bark had lower affinity for Zn than those from latrine waste and sewage sludge pyrolysed at 350 °C, both when the multi-metal effluent (Figure 1a) and single metal solution (Figure 1b) were used. Sorption maxima of Zn was higher on latrine waste and sewage sludge than pine-bark biochar, both for multiple metal effluent and single metal solution (Figure 2a,b). The sorption maximum values for all biochar types were lower for single metal solutions than multiple metal effluent. Biochar from latrine waste and sewage sludge pyrolysed at 550 and 650 °C exhibited lower affinity for Zn from multi-metal effluent than the same feedstocks pyrolysed at 350 °C. Pyrolysis at 650 °C resulted in lower Zn affinity for latrine waste biochar and higher affinity for sewage sludge biochar than biochar pyrolysed at 550 °C (Figure 1c). The maximum sorption of Zn on latrine waste biochar declined from 312.5 to 192.3 mg kg^−1^ with increasing pyrolysis temperature from 350 to 550 °C, with no further change at 650 °C (Figure 2b). Increasing pyrolysis temperature did affect Zn sorption on sewage sludge biochar. The constant related to bonding energy was higher on pine-bark than faecal biochar and was also higher for single metal solutions than the effluent (except latrine waste) (Table 1). While maximum sorption values were similar between latrine waste and sewage sludge biochar, the bonding energy was higher for sewage sludge biochar. Coefficients of variation for faecal biochar types (350 °C) were >0.90 and was 0.68 for pine-bark biochar from multiple metal effluent. However, constants related to bonding energy declined with increase in pyrolysis temperature for both latrine waste and sewage sludge biochar. Coefficients of variation were lower than 0.90 for at the 550 °C pyrolysis temperature, for both latrine waste and sewage sludge biochar.

### 2.2. Sorption of Cu from Multiple Metal Effluent and Single Metal Solutions

Similarly to Zn, the shape of the sorption isotherms shows that pine-bark biochar had lower affinity for Cu than those from latrine waste and sewage sludge pyrolysed at 350 °C, both for the effluent (Figure 3a) and single metal solution (Figure 3b). Affinity for Cu from single metal solution was higher for latrine waste biochar than sewage sludge biochar, while for the effluent sewage sludge biochar had greater affinity. The sorption maxima of Cu from single metal solution was higher for latrine waste and sewage sludge biochar and lower for pine-bark biochar when compared with those of multi-metal effluent (Figure 4a). Also similar to Zn, maximum sorption of Cu was higher for biochar from latrine waste and sewage sludge than pine-bark biochar, both for the effluent and single metal solution. Sorption of Cu from multi-metal effluent was lower on latrine waste biochar than sewage sludge, while for single metal solution, sorption on latrine waste biochar was higher (149.6 mg kg^−1^) than on sewage sludge (128.4 mg kg^−1^). Also similar to Zn, the constant related to bonding energy was higher for single metal solutions than the effluent, except pine-bark biochar (Table 2). Latrine waste biochar had the highest bonding energy constant while pine-bark biochar had the lowest. All coefficients of variation were >0.95, except for Cu sorption on latrine waste biochar from single metal solution.

Maximum sorption of Cu from effluent decreased with increasing pyrolysis temperature for the faecal biochar types, with 128 mg kg^−1^ at 350 °C and 40.3 mg kg^−1^ at 650 °C sewage sludge biochar (Figure 4b). On latrine waste biochar, Cu sorption also decreased with increasing pyrolysis temperature from 150 mg kg^−1^ at 350 °C to 72.9 mg kg^−1^ at 650 °C. The *b*-values also decreased with increase in pyrolysis temperature (Table 2). The only coefficients of variation that were <0.9 were those on biochar produced at 650 °C. Pine-bark 550 and 650 °C were not tested for metal sorption following their low metal sorption affinities at 350 °C.

### 2.3. Sorption of Cr from Multiple Metal Effluent and Single Metal Solutions by Biochar

The adsorption isotherms for Cr on latrine waste, sewage sludge and pine-bark biochar types are shown in Figure 5. The isotherms fitted well in the Langmuir model. The maximum sorption of Cr was higher (at least eight times) for single metal solutions than multiple metal effluent for all biochars (Figure 6). Sorption maxima of Cr were in the order pine-bark > latrine waste > sewage sludge, both for single metal and multiple metal effluent solutions. The constant related to bonding energy was lower for single metal solutions than the effluent for all biochar types pyrolysed at 350 °C. The coefficient of bonding energy was highest for pine-bark followed by latrine waste. On pine bark biochar pyrolysed at 550 and 650 °C, the isotherms suggest that all the Cr was removed (sorbed), leaving no Cr in solution, as such a sorption maximum could not be calculated because of lack of fit to the Langmuir isotherm (Figure 5c). On sewage sludge, Cr sorption declined with increase in pyrolysis temperature, with the highest bonding energy coefficient (79 L kg^−1^) for the 350 °C biochar, and lower (1.0 L kg^−1^) for higher temperatures. Again, Cr sorption by latrine waste at 350 °C from the effluent did not fit Langmuir model (Table 3).

### 2.4. SEM-EDS Chemical Composition of Biochars with Sorbed Metals

Major elements from the EDS spectra are tabulated in Table 4 with C and O being dominant for all biochar types. Molar O:C ratios of the biochars were highest on latrine waste and least on pine-bark biochar. Where sorption was done with effluent (50% dilution), the concentrations of Zn, Cu and Cr were higher than in the control for latrine waste and sewage sludge biochar, with only Cr detected on pine-bark biochar.

### 2.5. Regression Analysis on Parameters Affecting Sorption of Metals from Effluents

Best-fit models of stepwise regression analysis results for Zn, Cu and Cr sorption and physicochemical properties of different biochars are presented in Table 5. Volatile matter, ash and fixed C contents and pH are the most influential biochar parameters on Zn sorption on biochars used in this study. These predictor parameters were significantly influential (at 95% level) for the best-fit model which explained 99% of the variation. The model indicates that sorption maximum of Zn is likely to increase with increase in volatile, ash, fixed C and pH. The best-fit model from step-wise regression revealed that the combined effects of ash and fixed C content of the biochar significantly influenced Cu sorption on the biochars. The model, which explained 94% of variation of Cu sorption on these biochars, reveals that copper sorption on these biochars is decreased if ash and fixed C increase. The computed stepwise regression analysis for Cr sorption showed the highly significant influence of ash content and BET surface area. An increase in these two predictor variables decrease Cr sorption, with the two variables explaining 96% of variation Cr sorption on these biochar types.

## 3. Discussion

The higher Zn and Cu sorption maxima of biochar from latrine waste and sewage sludge than pine-bark biochar, both for multiple metal effluent and single metal solution, could be related to biochar characteristics as affected by feedstock. Faecal biochar had higher ash, P content and pH than pine-bark biochar, which could have supported precipitation. Zhou et al. [30] reported that peanut-shell biochar with ash had higher Cu sorption than where the ash was removed. The relationship of Zn sorption with ash content of biochar was also in agreement with [31], who reported that higher ash released hydroxide and carbonates, which formed precipitates with Zn. Hegazi [32] also reported that effective removal of metals, including, Cu was increased by higher ash content, with precipitation of the metals on biochar occurring due to phosphates, carbonates and hydroxides [24,30]. This view was supported by the results of the stepwise regression analysis, which showed positive effects of pH and ash, in addition to volatile matter and fixed C. The higher ash and pH, together with higher P, in the latrine waste and sewage sludge biochar types used in this study, could, therefore, have facilitated precipitation of Zn and Cu. However, stepwise regression analysis showed that Cu sorption was reduced by higher ash and fixed C contents, while Cr was reduced by ash and specific surface area, suggesting contributions of different mechanisms between all three metals. Although sewage sludge biochar had lower ash than that from latrine waste, the higher P could have contributed in greater precipitation. The trend of Cu sorption on the biochar types was in agreement with [19], who reported that dairy manure biochar was more efficient in removing Cu than that from plant (rice husk) material. The sorbed Cu and Zn were supported by results of SEM-EDS, where the elements were higher than in the control. The ash content of pine-bark biochar was too low to cause significant precipitation of Zn and Cu sorption. Based on these results, biochar from sewage sludge would be more effective as a sorbent to remove Zn and/or Cu from multi-metal effluents where any of these metals are dominant, while latrine waste biochar would be more effective for single metal solutions.

The higher Cr sorption capacity of biochar from pine-bark than from sewage sludge, in single metal solutions, suggests that a different mechanism from those of Zn and Cu, on this biochar. This trend was similar to that of surface area and cation exchange capacity (CEC). The higher surface area and CEC could have enhanced retention of Cr cations of the pine-bark biochar through ion exchange mechanism. Ding et al. [33] reported that hydroxyl and methyl functional groups were essential for the sorption and reduction of Cr(VI) to Cr(III), which precipitates with phosphate. Complexation, an important mechanism for Cr sorption [34], depends on the abundance of these functional groups, which in this research, declined with pyrolysis temperature on pine-bark biochar. The isotherms suggests that all the Cr was retained on latrine waste biochar at higher pyrolysis temperatures and this suggest that these important chemical functional groups (hydroxyl and methyl) were effective [35,36], as indicated by the decline in atomic ratio of H/C. The results of Cr sorption suggested that, in effluents where Cr is the dominant pollutant, pine-bark biochar would be more appropriate as a sorbent than biochar from latrine waste and sewage sludge.

The higher Zn sorption on all biochar types produced at 350 °C in multi-metal than single metal system was contrary to some findings in the literature [26]. Although [26] reported higher sorption of Cu than Zn in binary solution with equal concentrations of the two metals, competition with Zn at 20 times higher concentration, could have suppressed Cu sorption. Metal concentration play a significant role in the rate of adsorption [37]. The higher Zn and lower Cu and Cr sorption, in the multi-metal than single metal solutions, indicated that competitive sorption of the metals was affected by relative concentrations in the effluent, which favoured Zn sorption. The extremely higher Zn in the effluent than Cu and Cr could have precipitated with the hydroxides, carbonates and phosphates from the ash at the expense of Cu and Cr, which remained in solution. Competitive sorption of metals by different biochars has been reported previously [26]. The acidic nature of the effluent could also have lowered the CEC of the materials, which together with lower concentrations (relative to Zn) lowered Cu and Cr sorption, when compared to single metal solutions. Park et al. [38] indicated that single-metal system of Zn resulted in 38.6 mg g^−1^ sorption while binary-metal system, with equal concentrations with Cu, yielded 7.9 mg Zn g^−1^ on rice straw biochar. Zhou et al. [36] also reported that Zn sorption on sewage sludge biochar was 5.91 and 2.48 mg g^−1^ in mono-metal and multi-metal solution, respectively.

In the current study, the lower Cu sorption values in single metal solution than in multiple metal effluent, except for pine-bark, could be attributable to high Zn concentration and competition, which was higher than the other elements. Mishra et al. [37] reported that adsorbent dosage and metal concentration play a significant role in the rate and quantity of adsorption [37]. Park et al. [38] also reported lower values of Cu sorption in multi-metal (40.2 mg g^−1^) than single metal solutions (56.5 mg g^−1^) due to competitive sorption with co-existing metals [34] especially Zn. The lower sorption of Cr from multi-metal than single metal model could be explained by competition for sorption sites, with Zn and Cu, which had 170 and eight times higher concentration, respectively. In studies comparing single and competitive metal sorption, Cr was found to be easily exchanged and substituted by other metals such as Cu and Zn [26,39]. Since Cr sorption also follows the trend of surface area and CEC than the other two metals, the acidic effluent could have lowered the CEC of the biochars, lowering the retention of Cr when compared with single metal solutions.

The lack of change of Zn sorption and lower Cu sorption maximum on sewage sludge with increasing pyrolysis temperature could be explained by the increase in P in the sewage sludge biochar. The extremely high concentration of Zn could have precipitated with the increased P in the sewage sludge biochar, suppressing Cu sorption on the same biochar. The decrease in Cu sorption on sewage sludge biochar, with increasing pyrolysis temperature could be due to loss of oxygen-containing functional group such C=O stretching of aldehyde and carbonate ions at 550 and 650 °C [28]. The reason for the decline of Zn sorption and no change in Cu sorption on latrine waste with increasing pyrolysis temperature was not clear. However, the pH of the equilibrium solution could have contributed. Other studies reported that adsorption capacity for Cu (II) by biochar was increased with increase in pH from 2.0 to 6.0 especially on Fe and Zn laden biochar [36]. However, [31] reported that biochar produced from rice straw at low pyrolysis temperature (300 and 400 °C) showed the higher removal capacity of metals than those at higher pyrolysis temperatures. Zinc sorption may decrease with increase in pyrolysis temperature owing to loss of chemical functional groups that offer site for physical adsorption. Increasing pyrolysis temperature would have formed less reactive biochar rich in condensed C.

Several other studies reported that pyrolysis temperature influences sorption capacities of different biochars [17,24,31]. For example, Cr (III) removal by low pyrolysis temperature biochar (100 and 300 °C) was higher than high pyrolysis temperature (500 and 700 °C) [31]. On the other hand, removal of Cu, Zn and Cd by 350 °C dairy manure biochar was more efficient than 200 °C biochar of the same feedstock [24]. The current study suggested that, at lower pyrolysis temperature, sewage sludge and latrine waste biochar types were more effective than pine-bark biochar, while at higher temperature, sewage sludge was better that latrine waste biochar for sorption of Zn.

Bonding energy is a key quantity describing the strength of the interaction of molecules with the surface [40]. The bonding energy for Zn was lower on pine-bark than on faecal waste derived biochars. The strength of the binding between a given metal and a surface may vary from reaction site to the other [41]. It may be because each energy represent the contribution made by the bond to the total atomization energy of the concerned molecule [42]. Hence, at the greater adsorption density (ions/m^2^ specific surface area), the strongest binding site becomes limiting and average binding constant decreases even though the total number of available unoccupied binding site is large [41]. The lower bonding energy constant of Cu for effluent than single metal solution suggested that sorbed Cu could easily be desorbed. On the other hand, the higher coefficient of bonding energy of Cr on for pine-bark biochar than latrine waste and sewage sludge biochar, suggests stronger bonding, possibly minimising the possibility of desorption. This further supported the view that a different sorption mechanism could have been involved. The lower bonding energy constants of Cr for single metal solutions than effluent for all biochar types at 350 °C, suggested weaker bonding, associated with easier desorption.

## 4. Materials and Methods

### 4.1. Biochar

The biochar types used in this study were produced from two faecal wastes (sewage sludge and latrine waste) and pine-bark. The preparation of the biochars from the different feedstocks was as detailed in [29]. Briefly, it involved drying at 70 °C for 24 h and milling (<5 mm) before slow pyrolysis for 2 h in a muffle furnace, under limited oxygen at 350, 550 and 650 °C [43,44]. The biochar was weighed before characterization [45] and selected characteristics of the biochars are shown in Table 6 and Table 7.

### 4.2. Industrial Effluent

Untreated industrial effluent was obtained from ES & LC Manufacturing (Pietermaritzburg, South Africa), an electroplating company in Pietermaritzburg (29°39′03.03″S and 30°24′43.53″E). The company specializes in making and electroplating satellite mounting brackets. The effluent was analysed for pH, electric conductivity (EC), and chemical oxygen demand (COD), according to standard methods [46]. Total elemental composition of the effluent was analysed using the inductively coupled plasma-optical emission spectrometry (ICP-OES, Varian 720-ES series, by Varian Australia Pty Ltd) [47]. This effluent had low pH, high electrical conductivity and a variety of heavy metals, as indicated in Table 8. Three metals had concentrations higher than the critical limits for disposal into the environment and were in the order Zn > Cu > Cr.

### 4.3. Sorption of Elements from Single Metal Solutions and Effluent with Multiple Metals by Biochars

The batch sorption study with biochar from latrine waste, sewage sludge and pine-bark biochar, pyrolysed at 350 °C, was done with Zn solutions of increasing concentrations. An analytical grade stock solution of 1000 mg Zn L^−1^ was diluted in a 0.01 M CaCl_2_ as a background electrolyte. The solution was diluted to 5, 10, 15, 20, and 25 mg Zn L^−1^ solutions with 0.01 M CaCl_2_. The 0.01 M CaCl_2_ solution was used as the 0 mg L^−1^ concentration. The batch tests were replicated three times and blanks were run concurrently [8]. The Zn solutions (25 mL) were added to centrifuge tubes containing 2.5 g biochar samples. The suspensions were shaken at 180 r min^−1^ at constant temperature (25 °C) for 24 h [49] on a reciprocating shaker. The suspensions were centrifuged at 9440× *g* for 10 min and the supernatants filtered through Whatman No. 42 filter paper [49]. A Fast Sequential Atomic Absorption Spectrometer (Model AA280FS) was used to measure Zn concentration in the supernatant. Amount of Zn sorbed (*S*) was calculated as the difference between amount added and amount in the equilibrium solution, using Equation (1) [50]:(1)$$S=\frac{\left(C_{0}-C_{e}\right)\;\times \;V}{W}$$
where *S* is the amount of Zn sorbed (mg kg^−1^), *C*_0_ and *C*_e_ are the initial and equilibrium solution Zn concentrations (mg L^−1^), *V* is the solution volume (L), and *W* is the adsorbent weight (kg).

The same batch sorption study was repeated with solutions of Cu and Cr, at the same concentrations as Zn. Only the pH of metal solutions with increasing concentrations were measured and reported as shown in Table 9.

Another batch sorption study with the same biochars was conducted with dilutions of the industrial effluent. The effluent was diluted to the selected metal concentrations with 0.01 M CaCl_2_ solution (background electrolyte). Biochar (2.5 g) was mixed with 25 mL of 0%, 6.25%, 12.5% and 25% for Zn and 0%, 12.5%, 25%, 50%, 75% and 100% of the effluent for Cu and Cr. These dilutions corresponded to 0, 13.3, 26.5 and 53.0 mg L^−1^ for Zn; 0, 0.638, 1.28, 2.55, 5.10, 7.65 and 10.2 mg L^−1^ for Cu; and 0, 0.078, 0.156, 0.313, 0.625, 0.938 and 1.25 mg Cr L^−1^. The batch sorption study with the effluent was repeated with biochar produced from latrine waste and sewage sludge pyrolysed at 550 and 650 °C to determine effects of pyrolysis temperature. Pine-bark biochars were not used because they showed low Cd sorption in the preliminary study, and when pyrolysed at 350 °C, Zn and Cu sorbed was low. To estimate the sorption capacity and parameter related to energy of bonding of heavy metals onto biochar, the experimental data were fitted to the Langmuir sorption model (Equation (2)), from which the *S*_max_ and *b*–values were calculated [24,51].
(2)$$\frac{C}{S}=\frac{1}{S_{\max}\;\times \;b}+\frac{C}{S_{\max}}$$
where *C* is equilibrium concentration; *S*, amount of metal sorbed; *S*_max_ was maximum amount of metal that the biochar can sorb; and *b* is constant related to binding affinity.

The solid retained on the filter paper was collected for Scanning Electron Microscopy-Energy Dispersive X-ray Spectroscopy (SEM-EDS) to determine abundance of specific available and adsorbed elements [24].

### 4.4. Surface Characteristics of Resultant Biochars

The ZEISS EVO LS15 scanning electron microscopy-energy dispersive X-ray spectroscopy (SEM, Oberkochen, Germany) was used to analyse the external morphology (surface characteristics) and qualitative chemical composition of the biochars. The analyses involved a beam of electrons generated in a vacuum, which is collimated by electromagnetic condenser lenses and scanned across the sample surface by a coil. Secondary electrons were then made to fall on the surface of a photosensitive plate in a photomultiplier tube. Amplified electrons are send to phosphorescent screen which provided magnified image of sample surface. Energy Dispersive X-ray Spectroscopy (EDS) was used to analyse the energy spectrum in order to determine abundance of specific available elements in a given sample. Figure 7 shows distinctive morphological structure of the 350 °C biochars. External morphology of the resultant biochars shows porous structure with some fissures and or flakes, which varied between biochars from the different feedstocks.

### 4.5. Statistical Analysis

Stepwise regression derived from multiple regression analysis was used to determine the combined effect of biochar parameters on metal sorption capacity from the industrial effluent. The analysis identified the best-fit model by removing certain variables based on the t-statistics for estimated coefficients [52].

## 5. Conclusions

Biochar types from sewage sludge and latrine waste removed more Zn and Cu and less so for Cr, from multiple metal effluent and single metal solutions, than pine-bark biochar. Sorption on the biochars was higher for Zn and lower for Cu and Cr, from multi-metal effluent than from single metal solutions. Increase in pyrolysis temperature reduced sorption of Zn, Cu and Cr from multi-metal effluent on latrine waste and sewage sludge biochar. Pine-bark biochar (350 °C) had higher Cr sorption than its counterpart latrine waste and sewage sludge both for multi-metal and single metal solution. Biochar derived from human faecal waste, pyrolysed at 350 °C promises to be an effective sorbent for removal of Zn, Cu and Cr from solution, with greater sorption of Zn from multi-metal effluent on sewage sludge biochar, and from single metal solution on latrine waste biochar. The use of these feedstocks for biochar production from the stock-piles could alleviate their disposal challenges. The cost-effectiveness of the biochars will depend on the costs of drying the feedstocks and pyrolysis and the disposal strategies for the metal enriched biochars, which still need to be studied.

## Figures and Tables

**Figure 1 materials-12-01768-f001:**
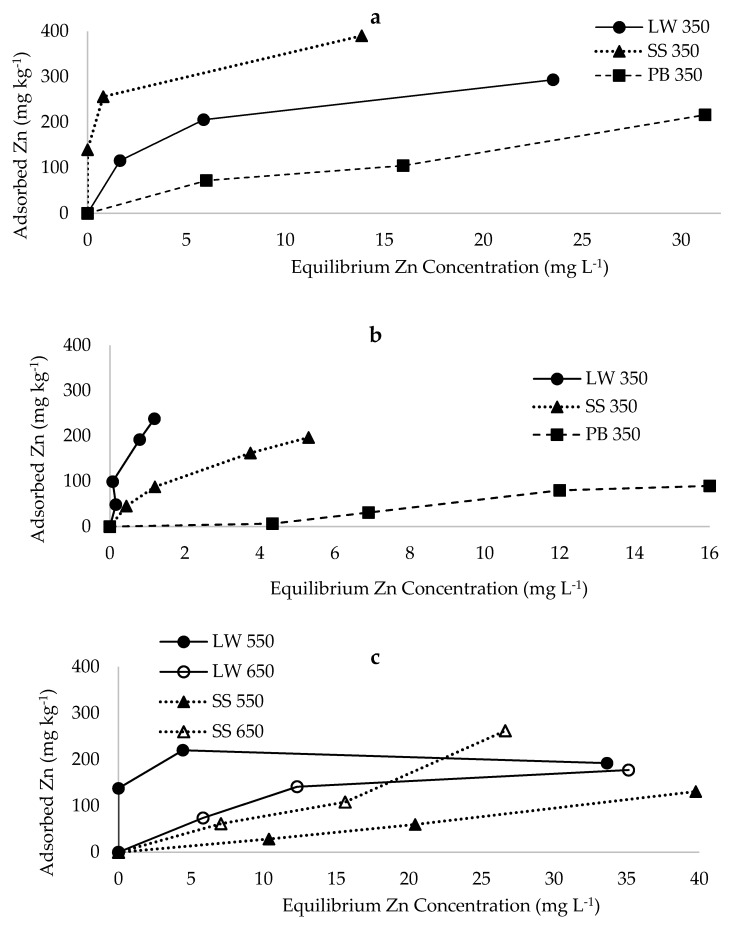
Sorption isotherms of zinc: (**a**) from effluent on biochar pyrolysed at 350 °C, (**b**) single metal on biochar pyrolysed at 350 °C and (**c**) from effluent on biochar pyrolysed at 550 and 650 °C. LW = latrine waste biochar, SS = sewage sludge biochar, PB = pine-bark biochar.

**Figure 2 materials-12-01768-f002:**
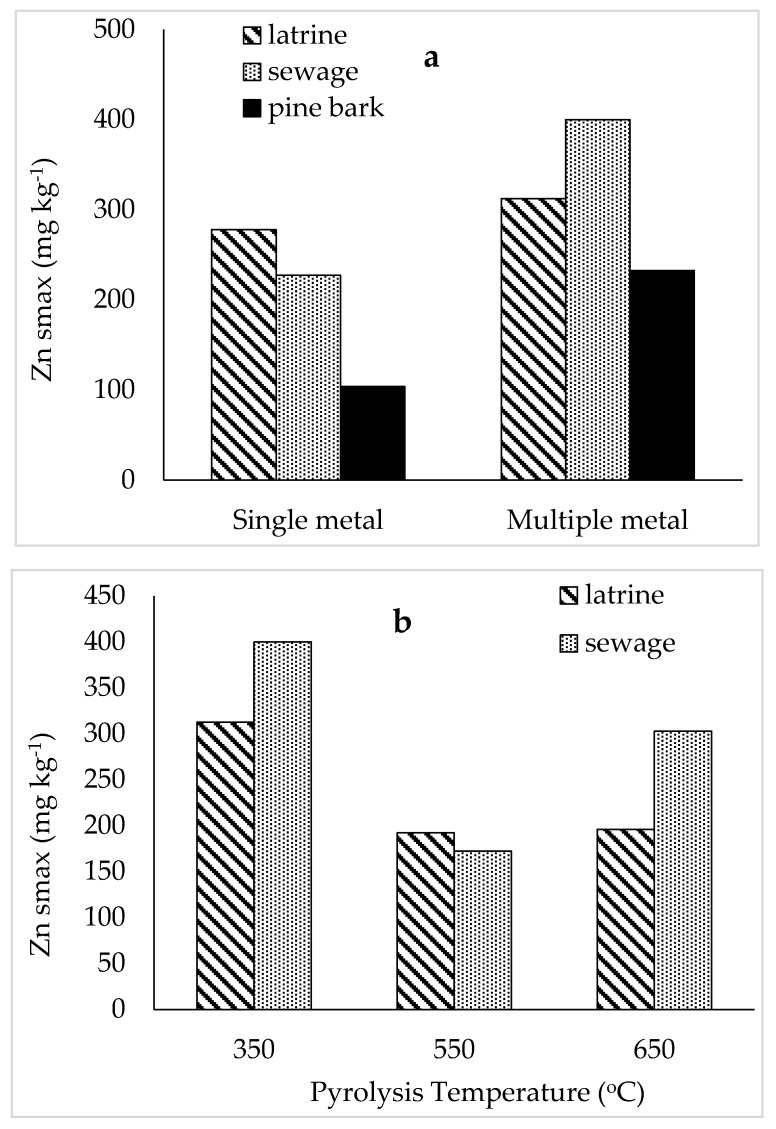
Maximum zinc sorption on faecal and pine-bark biochar pyrolysed at 350 °C (**a**) and on faecal biochar at increasing pyrolysis temperatures (**b**).

**Figure 3 materials-12-01768-f003:**
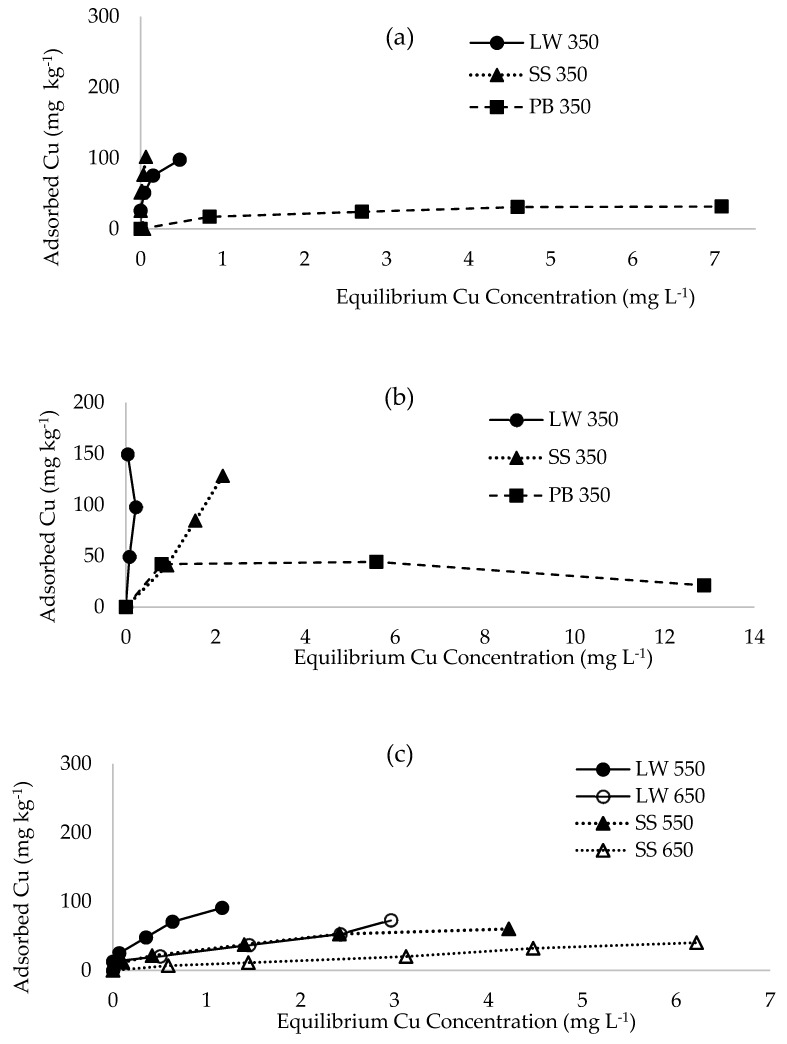
Sorption isotherms of copper from: (**a**) effluent on biochar pyrolysed at 350 °C, (**b**) single metal on biochar pyrolysed at 350 °C and (**c**) effluent on biochar pyrolysed at 550 and 650 °C. LW = latrine waste biochar, SS = sewage sludge biochar, PB = pine-bark biochar.

**Figure 4 materials-12-01768-f004:**
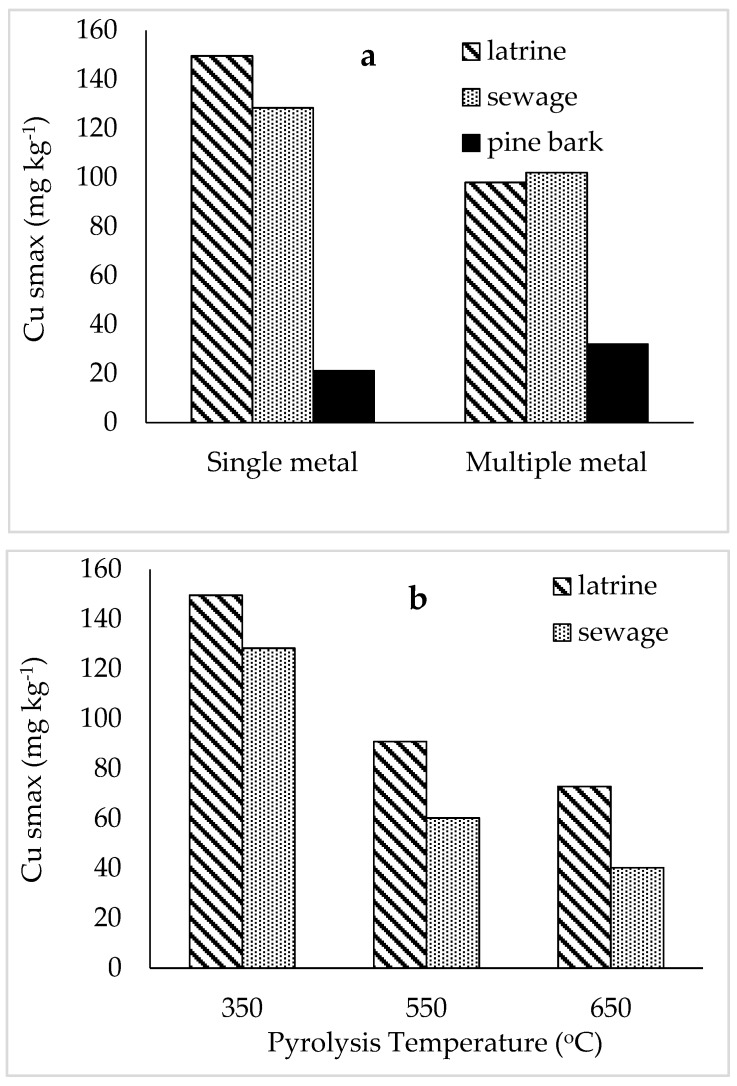
Maximum copper sorption on faecal and pine-bark biochar pyrolysed at 350 °C (**a**) and on faecal biochar at increasing pyrolysis temperatures (**b**).

**Figure 5 materials-12-01768-f005:**
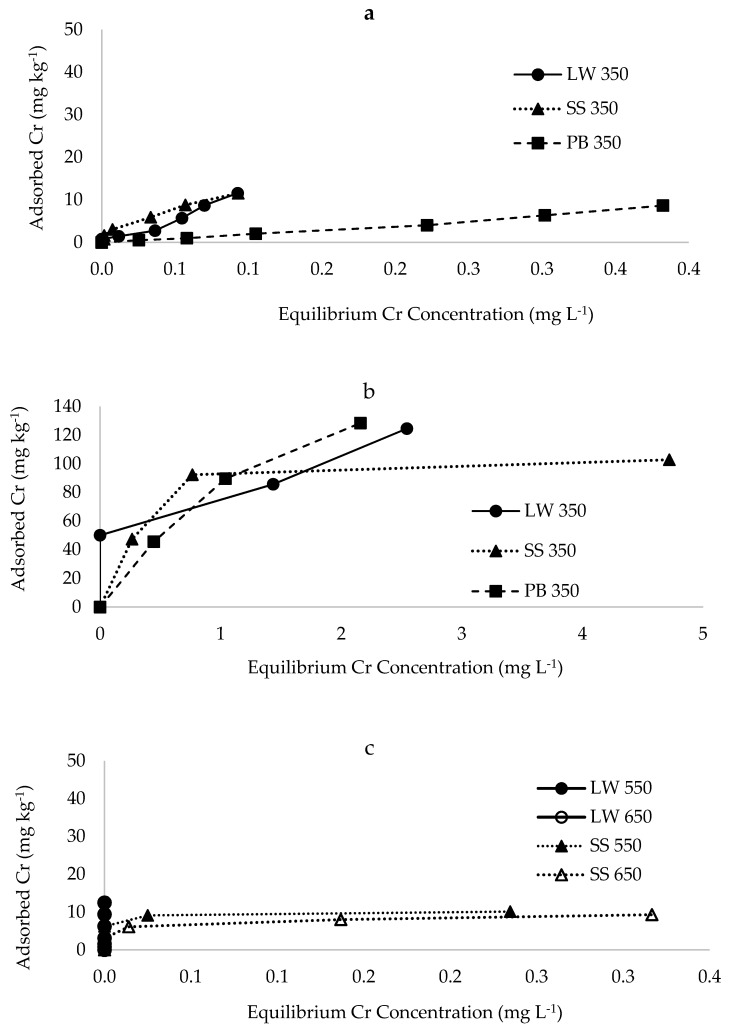
Sorption isotherms of chromium from: (**a**) effluent on biochar pyrolysed at 350 °C, (**b**) single metal on biochar pyrolysed at 350 °C and (**c**) effluent on biochar pyrolysed at 550 and 650 °C. LW = latrine waste biochar, SS = sewage sludge biochar, PB = pine-bark biochar.

**Figure 6 materials-12-01768-f006:**
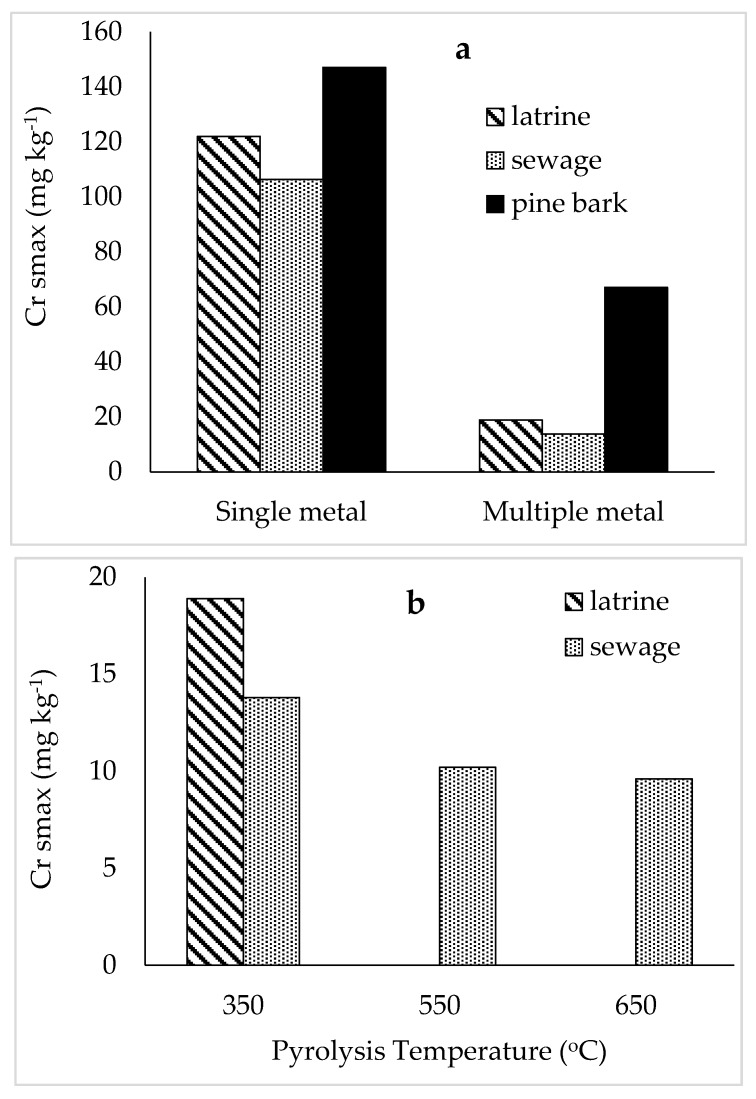
Maximum chromium sorption on faecal and pine-bark biochar pyrolysed at 350 °C (**a**) and on faecal biochar at increasing pyrolysis temperatures (**b**).

**Figure 7 materials-12-01768-f007:**
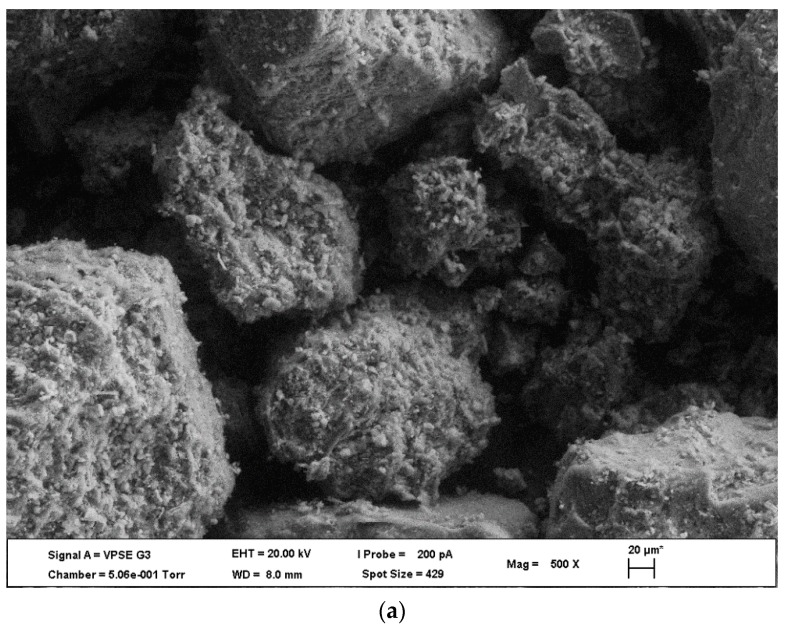
Morphological comparison of biochar types produced at 350 °C from (**a**) latrine waste, (**b**) sewage sludge and (**c**) pine bark.

**Table 1 materials-12-01768-t001:** The *b* (L kg^−1^) and *r*^2^ values of linearized Langmuir models of zinc on biochars from latrine waste, sewage sludge and pine bark.

Biochar	*b*-Value and *r*^2^			
	Source of Metals (350 °C Pyrolysis)		Pyrolysis Temperature (Metals from Effluent)	
	Aqueous Solution	Effluent	550 °C	650 °C
Latrine waste	3.6 (0.77)	0.55 (0.98)	−3.71 (1.00)	0.22 (0.94)
Sewage sludge	0.86 (0.90)	5.00 (1.00)	0.04 (0.34)	0.07 (0.38)
Pine-bark	0.18 (0.48)	0.11 (0.68)	–	–

**Table 2 materials-12-01768-t002:** The *b* (L kg^−1^) and *r*^2^ values of linearized Langmuir models of copper on biochars from latrine waste, sewage sludge and pine bark.

Biochar	*b*-Value and *r*^2^			
	Source of Metals (350 °C Pyrolysis)		Pyrolysis Temperature (Metals from Effluent)	
	Aqueous Solution	Effluent	550 °C	650 °C
Latrine waste	51.5 (0.84)	32.7 (0.99)	5.67 (0.92)	1.15 (0.75)
Sewage sludge	1.22 (0.46)	510 (0.97)	1.82 (0.95)	0.32 (0.58)
Pine-bark	−1.25 (0.95)	1.97 (0.98)	–	–

**Table 3 materials-12-01768-t003:** The *b* (L kg^−1^) and *r*^2^ values of linearized Langmuir models of chromium on biochar from latrine waste, sewage sludge and pine bark.

Biochar Feedstock	*b*-Value and (*r*^2^)			
	Source of Metals (350 °C Pyrolysis)		Pyrolysis Temperature (Metals from Effluent)	
	Aqueous Solution	Effluent	550 °C	650 °C
Latrine waste	5.47 (0.92)	9.64 (0.19)	–	–
Sewage sludge	6.71 (1.00)	38.3 (0.92)	980 (1.00)	64.8 (1.00)
Pine-bark	2.06 (0.83)	0.29 (0.31)	–	–

* No Cr sorption occurred on latrine waste biochar at 550 and 650 °C.

**Table 4 materials-12-01768-t004:** Concentrations of carbon, oxygen, copper, chromium and zinc (%) on biochar types pyrolysed at 350 °C after sorption study with or without effluent.

Solution Used for Sorption	Element (weight %)	Biochars		
		Pine-Bark	Sewage Sludge	Latrine Waste
**No effluent**	C	71.9	60.9	*34.0*
	O	28.1	29.0	*42.7*
	O:C	0.29	0.36	0.94
	Cu	*	0.01	*
	Cr	*	0.06	*
	Zn	*	0.05	*
50% effluent	C	74.6	61.7	43.9
	O	25.4	28.8	40.8
	O:C	0.26	0.35	0.70
	Cu	*	0.07	0.09
	Cr	0.03	0.08	0.18
	Zn	*	0.21	0.09

***** indicate non-detection of the metals.

**Table 5 materials-12-01768-t005:** Step-wise regression analysis for parameters affecting Zn, Cu and Cr sorption maximum: presentation of the best-fit models.

Predictor Variable	Parameter Estimates	Standard Error	*r* ^2^
**Zinc**			
Intercept	−37760.85	7293.96	
Volatile (%)	439.08 *	84.09	
Ash (%)	299.51 *	64.13	
Fixed C (%)	259.64 *	57.91	
pH	1094.26 *	122.69	0.9889
**Copper**			
Intercept	535.73	62.59	
Ash (%)	−4.49 **	0.65	
Fixed C (%)	−8.36 **	1.08	0.9381
**Chromium**			
Intercept	21.32	1.83	
Ash (%)	−0.17 **	0.02	
BET (m^2^ g^−1^)	−0.25 ***	0.03	0.9604

*, **, *** Indicates the significance at the 95%, 99% and 99.9% level, respectively.

**Table 6 materials-12-01768-t006:** Selected physicochemical properties of biochars used in this study.

Biochar	pH (KC)	Ash (%)	Fixed C (%)	Volatile (%)	SSA (mg^−1^)	PV (cm^3^ g^−1^)	Pore Size (Å)
Latrine Waste							
350	6.9	84.3	5.5	10.27	7.54	0.035	183
550	7.1	89.9	4.4	5.70	23.7	0.053	91.0
650	7.3	92.7	4.2	3.10	25.7	0.052	81.2
Sewage Sludge							
350	6.0	49.2	14.6	31.13	0.29	0.0008	113
550	7.7	66.2	23.9	9.80	1.58	0.0051	130
650	8.0	69.4	23.3	7.23	4.21	0.0073	50.6
Pine-Bark							
350	4.3	–	57.6	39.47	73.1	0.053	28.8

* Biochars from pine-bark produced at 550 and 650 °C were not used in this study. SSA, PV and Pore size represent BET specific surface area, pore volume and average pore size, respectively.

**Table 7 materials-12-01768-t007:** Total composition of selected elements and CEC in biochars used in this study.

Biochar	N (%)	P (g kg^−1^)	K	Ca	Mg	Na	CEC (cmolc kg^−1^)
			(mg kg^−1^)				
Latrine Waste							
350	1.04	25.3	368	1953	374	207	5.09
550	0.71	24.9	407	3469	493	322	4.91
650	0.44	25.0	227	3769	525	225	5.65
Sewage Sludge							
350	5.45	22.6	1290	1097	593	462	4.17
550	2.64	38.1	489	557	418	101	2.31
650	2.77	37.9	567	640	565	235	2.40
Pine-Bark							
350	0.69	2.6	1780	1200	280	410	15.0

* Biochars from pine-bark produced at 550 and 650 °C were not used in this study. CEC represents cation exchange capacity.

**Table 8 materials-12-01768-t008:** Electrical conductivity, pH and concentrations of heavy metals in the effluent used.

Parameters	Critical Limit (DWAF, 2004 In [48])	Effluent
pH	5.5–9.5	2.5 ± 0.0
EC (mS m^−1^)	70–150	1383 ± 4.2
COD (mg O_2_ L^−1^)	<75	510 ± 36.8
Cr (µg L^−1^)	<0.05	1.25 ± 0.02
Cu (mg L^−1^)	<0.01	10.2 ± 0.24
Zn (mg L^−1^)	<0.10	212 ± 13.8

Critical limits (DWAF, 2004). Metal ratios: Zn:Cu = 21:1, Zn:Cr = 170:1, Cu:Cr = 8:1.

**Table 9 materials-12-01768-t009:** pH of single metal solutions used in the study.

Metal Concentration	Cu	Cr	Zn
0 ppm	6.11	6.09	6.09
5 ppm	2.83	5.54	2.84
10 ppm	2.54	5.50	2.56
15 ppm	2.31	5.43	2.40
20 ppm	2.22	5.39	2.25
25 ppm	1.96	5.27	2.17

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
