# Peer review of "Sorption of Selected Heavy Metals with Different Relative Concentrations in Industrial Effluent on Biochar from Human Faecal Products and Pine-Bark"

_materials, 2019, doi:10.3390/ma12111768_

Reviewer 1 Report

The submitted article investigates biochars form different feedstock and pyrolysis settings as adsorbent for Zn, Cu and Cr elements. The Authors applied single metal solutions and wastewater effluent with different dilutions. 

The single Cr and Cu adsorption experiments cover a much wider concentration range than the experiments with effluent.

There are no data about the pH changes during the batch experiments. The effluent is highly acidic (pH 2.5) which may have a significant effect on the surface charges of biochars that have neutral pH. We do not know the pH of the single metal solutions. The pH of equilibrium suspensions may be also important to analyse. Because of these pH differences it would be necessary to determine the effective and potential CEC of the applied biochars. Proper adsorption analysis without CEC values is not acceptable. The BET values were only marginally used in discussion either.

Materials and methods

The 4.3 section is very confused. It is hard to follow how the experiments were performed. Line 323, 326: there is a reference for „Section 3.2.6” however, there is no such section in the text. The details of the batch experiment should be described properly. What was the reason to make the 3rd batch experiment when its materials and treatments were already included in the 2nd batch experiment?

Why did the authors describe the batch experiments in less details again for the SEM-EDS method?

Line 358. According to that the authors made a stepwise regression to analyse the data. In the results and discussion it has not been mentioned. However, it would be interesting to see this evaluation.

The Langmuir model with defined parameters (used for Table 1-3) should be included in materials and methods.

Results

Line 92: How was effectiveness calculated?

SD values should be included in the figures.

Statistical analysis is necessary to see whether the differences between adsorption maximums are significant.

Uniform labels (Smax or sorption) of fig 2, 4, 6.

The decimals should be used uniformly. For values above 100 it is not necessary to use decimals.

The LW, SS, PB abbreviations should be defined.

Why include pine bark biochar 550 and 650 properties into Table 5 if it is not discussed?

The tables should be formatted properly.

Line 137: “Maximum sorption of Cu from effluent decreased with increasing pyrolysis temperature for the faecal biochar types, with 200 mg kg-1 at 350°C and 23 mg kg-1 at 650°C sewage sludge biochar.” Where did these values come from? They are different from those of Fig. 4.

Biochar CEC values and the pH of the suspensions should be included.

Discussion

The effect of pH on biochar CEC values should be considered in the discussion.

Conclusions

The Authors claim that the application of human faecal based biochar as a heavy metal adsorbent may alleviate the disposal problems of these materials. But is it feasible to dry and then make a biochar from these materials for effluent treatment? The costs of processing can be higher than the achieved benefit. And after their depletion the metal loaded biochars should be treated.

Author Response

The changes made in response to review comments are shown on the manuscript in red colour and are italicized for ease of tracking them.

Query: Line 289 -296: The section appears similar to published paper.

Response: The section was revised to read: “The preparation of the biochars from the different feedstocks was as detailed in Koetlisi and Muchaonyerwa (2018). Briefly, it involvd drying at 70°C for 24 h and milling (<5mm) before slow pyrolysis for 2 h in a muffle furnace, under limited oxygen at 350, 550 and 650°C (Yuan et al., 2011; Enders et al., 2012). The biochar was weighed before characterization (Herath et al., 2013) and selected characteristics of the biochars are shown in Tables 5 and 6”.

Appendix 1 was deleted as it was not of much relevance to the paper and it was not discussed.

REVIEWER 1

Materials and Methods

Query: The section 4.3 is very confused.  It is hard to follow how the experiments were performed. Lines 323, 326: There is a reference for section 3.2.6. However, there is no such section in the text. The details of the batch experiment should be described properly.

Response: The description was revised to readThe batch sorption study with biochar from latrine waste, sewage sludge and pine-bark biochar, pyrolysed at 350°C, was done with Zn solutions of increasing concentrations (0, 5, 10, 15, 20, and 25 mg L-1 in 0.01 M CaCl2). The batch tests were replicated three times and blanks were run concurrently (Mohan et al., 2007). An analytical grade stock solution of 1000 mg Zn L-1 was diluted in a 0.01 M CaCl2 as a background electrolyte. The solution was diluted to 5, 10, 15, 20, and 25 mg Zn L-1 solutions with 0.01 M CaCl2. The 0.01 M CaCl2 solution was used as the 0 mg L-1 concentration. The Zn solutions (25 ml) were added to centrifuge tubes containing 2.5 g biochar samples. The suspensions were shaken at 180 r min−1 at constant temperature (25°C) for 24 h (Khodaverdiloo and Samadi, 2011) on a reciprocating shaker. The suspensions were centrifuged at 9440*g for 10 min and the supernatants filtered through Whatman no. 42 filter paper (Khodaverdiloo and Samadi, 2011). A Fast Sequential Atomic Absorption Spectrometer (Model AA280FS) was used to measure Zn concentration in the supernatant. The pH of the supernatant was also measured for the single metal solutions only. Amount of Zn sorbed (S) was calculated as the difference between amount added and amount in the equilibrium solution, using Equation 1 (Desta, 2013):

                        Equation 3.1

Where S is the amount of Zn sorbed (mg kg-1), C0 and Ce are the initial and equilibrium solution Zn concentrations (mgL-1), V is the solution volume (L), and W is the adsorbent weight (kg)”.

Lines 320 – 323: was modified to read “The same batch sorption study was repeated with solutions of Cu and Cr, at the same concentrations as Zn. Only the pH of metal solutions with increasing concentrations were measured and reported as shown in Table 8.” Table 8 was added in line 340 as shown below

Table 8: pH of single metal solutions used in the study

Metal concentration

Cu

Cr

Zn

0ppm

6.11

6.09

6.09

5ppm

2.83

5.54

2.84

10ppm

2.54

5.50

2.56

15ppm

2.31

5.43

2.40

20ppm

2.22

5.39

2.25

25ppm

1.96

5.27

2.17

Query: What was the reason to make a third batch experiment when its materials and treatments were already included in the second batch experiment?

Response: Lines 324 - 326: The statement was revised to read “Another batch sorption study with the same biochars was conducted with dilutions of the industrial effluent.” This was done to remove reference to biochars at 550 and 650°C. The third biochar was then included to investigate effects of pyrolysis temperature on sorption of metals in the effluent.

Query: Why did the authors describe the batch experiments in less detail, again for SEM-EDS method?

Response: The little detail was provided in error. The paper was a chapter of a thesis which had related earlier chapters. As such in the thesis, it was written that way to avoid repetition. However, this should have been corrected before submission.

Query: Line 358: According to that the authors made a stepwise regression to analyse the data. In the results and discussion it has not been mentioned. However, it would be interesting to see this evaluation.

Response: Stepwise regression results have been confined to let detailed description and explanation surface.

Query: The Langmuir model with defined parameters (used in Tables 1-3) should be included in the Materials and Methods.

Response: The Langmuir model was added on Line 337 as below: “

       (Equation 2)

Where C is equilibrium concentration; S, amount of metal sorbed; Smax was maximum amount of metal that the biochar can sorb; and b is constant related to binding affinity”.

Results

Query: Line 92: How was effectiveness calculated?

Response: The following statement “Pine-bark biochar was about 100 times less effective on Zn sorption compared to faecal biochar. However, latrine waste biochar sorbed more Zn from single metal solution than from multi-metal effluent, than biochar from sewage sludge.” in lines 91 to 92 was deleted to avoid confusion

Query: Statistical analysis is necessary to see whether the differences between adsorption maxima are significant.

Response: The comment is acknowledged and appreciated. However, replication during sorption studies is done for each initial solute concentration, and therefore, the sorption studies were not repeated to produce three replicate isotherms for each sample. One isotherm was produced and each point on the isotherm was an average of three replicates. This therefore gave one value of Smax and one b-value from the linearized Langmuir model. As such statistical analysis is not possible.

Query: Figures 2,4 and 6: Uniform labels (Smax or sorption)

Response: The labels were corrected to “Zn smax”, “Cu smax”, and “Cr smax”

Query: The decimals should be used uniformly. For values above 100, it is not necessary to use decimals.

Response: The decimals were corrected in the abstract, and Appendix 1 (table). For numbers below 100, three significant figures were used.

Query: The LW, SS and PB abbreviations should be defined.

Response: The following definitions were added on captions of Figures, 1, 3 and 5: “LW = latrine waste biochar, SS = sewage sludge biochar, PB = pine-bark biochar”.

Query: Table 5: Why include pine bark biochar 550 and 650 properties if it is not discussed.

Response: Characteristics of pine bark biochars 550 and 650 were removed from Table 5, as recommended by the reviewer.

Query: The tables should be formatted properly.

Response: The tables have been formatted to improve the quality, including decimal points on the results and borders.

Query: Line 137: Maximum sorption of Cu from effluent decreased with increasing pyrolysis temperature for the fecal biochar types, with 200 mg kg-1 at 350oC and 23 mg kg-1 at 650oC for sewage sludge biochar.” Where did these values come from? They are different from those on Figure 4.

Response: The section (from line 137) was revised to read: “Maximum sorption of Cu from effluent decreased with increasing pyrolysis temperature for the faecal biochar types, with 128 mg kg-1 at 350°C and 40.3 mg kg-1 at 650°C sewage sludge biochar. On latrine waste biochar, Cu sorption also decreased with increasing pyrolysis temperature from 150 mg kg-1 at 350oC to 72.9 mg kg-1 at 650°C. The b-values also decreased with increase in pyrolysis temperature (Table 2)”.

Query: Biochar CEC values and pH of the suspensions should be included.

Response: The biochar CEC values were added to Table 6 (new table of elemental composition).

Discussion

Query: The effects of pH on biochar CEC values should be considered in the discussion.

Response: The following statements were added to the discussion:

 “Line 212: This trend was similar to that of surface area and CEC. The higher surface area and CEC could have enhanced retention of Cr cations of the pine-bark biochar through ion exchange mechanism.”

Line 232: The acidic nature of the effluent could also have lowered the CEC of the materials, which together with lower concentrations (relative to Zn) lowered Cu and Cr sorption, when compared to single metal solutions.

Line 246: Since Cr sorption to follows the trend of surface area and CEC than the other two metals, the acidic effluent could have lowered the CEC of the biochars, lowering the retention of Cr when compared with single metal solutions. 

Query: The BET values were marginally used in the discussion.

Response: Response: The following statements were added to the discussion:

 Line 212: This trend was similar to that of surface area and CEC. The higher surface area and CEC could have enhanced retention of Cr cations of the pine-bark biochar through ion exchange mechanism.

Line 246: Since Cr sorption to follows the trend of surface area and CEC than the other two metals, the acidic effluent could have lowered the CEC of the biochars, lowering the retention of Cr when compared with single metal solutions. 

Conclusion

Query: The authors claim that the application of human faecal based biochar as a heavy metal adsorbent may alleviate the disposal problems of these materials. But is it feasible to dry and then make biochar from these materials for effluent treatment. The costs of processing can be higher than the achieved benefits. And after their depletion, the metal loaded biochars should be treated.

Response:  The Conclusion has been revised to read: “Biochar types from sewage sludge and latrine waste removed more Zn and Cu and less so for Cr, from multiple metal effluent and single metal solutions, than pine-bark biochar. Sorption on the biochars was higher for Zn and lower for Cu and Cr, from multi-metal effluent than from single metal solutions. Increase in pyrolysis temperature reduced sorption of Zn, Cu and Cr from multi-metal effluent on latrine waste and sewage sludge biochar. Biochar derived from human faecal waste, pyrolysed at 350°C promises to be an effective sorbent for removal of Zn, Cu and Cr from solution, with greater sorption of Zn from multi-metal effluent on sewage sludge biochar, and from single metal solution on latrine waste biochar. The use of these feedstocks for biochar production from the stock-piles could alleviate their disposal challenges. The cost-effectiveness of the biochars will depend on the costs of drying the feedstocks and pyrolysis and the disposal strategies for the metal enriched biochars, which still need to be studied.”

Reviewer 2 Report

Reviewer finds the article can be considered for publication. However some additional experiments are suggested to compliment the results found in the manuscript. At present, current version of manuscript cannot adequately explain why different biochars under similar conditions behave that way. Hence, the reviewer suggests additional experiments to find more evidence to support the observed results:

1) elemental analysis of biochars beyond what is presented in the manuscript: (e.g: N, S, K, Na, Ca).

2. BET surface analysis of biochars investigated in this study. E.g, surface area, pore volume, pore size distribution etc

3) Morphology studies of biochar investigated in this study.

Minor suggestions:

*References in abstract is unnecessary.

*Abstract of the manuscript did not summarize the findings of the study, instead it looks more like summary of introduction

*Reference styles are not consistent with the journal

* Suggest materials and methods to be presented immediately after introduction than at the end the manuscript.

Author Response

Query: Elemental analysis of biochars beyond what is presented in this manuscript (N, S, K, Na, Ca)?

Response: A new table on the elemental composition of the biochars was added (new Table 6) and the original Table 6 is now Table 7.

Query: BET surface analysis of biochars investigated in this study e.g., surface area, pore volume, pore size distribution etc.

Response: The BET surface area was already on Table 5. The average pore sizes and total pore volumes were added to the Table as recommended by the reviewer.

Query: Morphology studies of biochars investigated in this study.

Response: A new Figure 7 has been added on the morphology of the biochar types produced at 350oC.

Query: References in the abstract is unnecessary.

Response: We acknowledge and agree with this comment. However, there are no references in the abstract.

Query: Abstract of the manuscript did not summarise the findings of the study, instead it looks more like a summary of the introduction.

Response: This comment is appreciated. However, when we look at our abstract, it shows what the paper is about with the results indicated. I am not sure where the misunderstanding may have originated from. We therefore did not change the abstract significantly.

Query:  Reference styles are not consistent with the journal.

Response: The referencing style has been revised to be consistent with the journal requirements.

Query: Suggest Materials and methods to be presented immediately after the introduction than at the end of the manuscript.

Response: The comment was appreciated as the arrangement is inconsistent with the sequencing of sections of papers in journals. However, the instructions to authors advises that the Materials and Methods comes after the discussion. As such the section was not moved to after the introduction. However, we request assistance from the editor on what should be done in such a situation. If it has to be moved, we agree, it can be moved.

Round  2

Reviewer 1 Report

Table 6, Table 7: The “*” indicating the footnote is missing from the table.

Line 466 “Gray et al., 1999” is not in the reference list.

Reviewer 2 Report

Authors have significantly improved the manuscript based on reviewers comments. The manuscript is recommended for publication as presented.